# Preparation and Performance of PAN–PAC Nanofibers by Electrospinning Process to Remove NOM from Water

**DOI:** 10.3390/ma14164426

**Published:** 2021-08-07

**Authors:** Beata Malczewska

**Affiliations:** Institute of Environmental Engineering, Wroclaw University of Environmental and Life Sciences, pl. Grunwaldzki 24, 50-363 Wroclaw, Poland; beata.malczewska@upwr.edu.pl

**Keywords:** PAC, AC, electrospun membrane, NOM removal, filtration

## Abstract

The technology based on electrospun membranes exhibits great potential in water treatment. This study presents experimental data involving the fabrication of nanofiber membranes with powdered activated carbon (PAC) and its application for the removal of natural organic matter. The fabricated membrane materials were characterized using various techniques. These include scanning electron microscopy (SEM), Fourier transform infrared spectroscopy (FTIR), and X-ray diffraction analysis. The incorporation of PAC nanoparticles influences the structure and physicochemical properties as well as the transport and separation characteristics of the produced membranes. The applicability of the fabricated carbon-based membrane was tested in the filtration experiments. The fabricated membrane is characterized by a high NOM removal efficiency of 79% in the filtration process. Further modification of the membrane composition may result in a further increase in the efficiency of removing contaminants from water.

## 1. Introduction

Carbon materials (CM) have been extensively employed for the adsorption application in water and wastewater technology. The interest in carbon materials is connected with their characteristics and, in particular, with their high specific surface area, mechanical, thermal, and chemical stability. Carbon-based materials, namely activated carbon, carbon nanotubes, and graphene, are presently considered the most versatile materials in water treatment [1,2,3,4]. Among all carbon-based materials, activated carbon (AC) is the one used most often. AC is available as powdered activated carbon (PAC) and Granular Activated Carbon (GAC). They contain charcoal-derived carbonaceous material, which is characterized by a high porosity structure, a large surface area that ranges from 500 to 3000 m^2^ g^−1^, as well as low cost and good charge-holding capacity [1,3]. In essence, the adsorption efficiency of AC depends on the specific surface area (SSA), pore size distribution (PSD), pore volume (PV), and the presence of surface functional groups [5]. Recently, AC modifications have been attracting more attention in attempts to increase its affinity to target contaminants in order to improve the efficiency of their removal [6]. Therefore, there are many approaches to the fabrication of various carbon-based materials by physical, chemical, and biological modifications of AC characteristics.

AC is often used for the removal of organic compounds, particularly those causing specific taste and odor. Additionally, AC adsorption proved to be effective for the removal of several types of inorganic and organic contaminants both as a stand-alone step and in combination with other conventional and advanced water and wastewater treatment systems [2,6,7].

The membrane technology was very successful in the removal of targeted contaminants. Depending on the pore size, it is possible to remove contaminants at an ionic level [8]. A broader application of membrane technology is limited by its susceptibility to fouling. Fouling of membranes by natural organic matter (NOM) is attributed to the adsorption of NOM in membrane pores and/or to the formation of a gel layer that covers the membrane surface [9,10]. The conventional NOM pretreatment processes, such as coagulation with metal-based coagulants or adsorption onto PAC, can capture NOM to some extent, but there is still a need for more efficient and economical methods. Several authors observed a significant improvement in membrane performance after coagulation [11,12,13,14]. Therefore, it is recommended to use membranes in integrated membrane systems.

Conventional methods can also release secondary pollutants to the environment, e.g., chlorine reacts with NOM in water and forms carcinogenic by-products [9]. Many modifications were thus implemented to improve the conventional process. One of them is the combination of PAC with low-pressure membranes, known as Hybrid Membranes [11]. The use of PAC before filtration was reported to reduce membrane fouling [9].

Recently, much more attention was given to the membranes prepared via the electrospinning technique. Electrospun nanofibrous membranes (ENMs) found applications in direct filtration and as a support layer for thin-film composites. The modification of the polymer structure and its chemical functionality can potentially facilitate the fabrication of highly selective membranes. ENMs combine both adsorption and filtration. Instead of being added to the feed water, the functional groups of adsorbents are combined with the surface and the pore wall of polymer membranes. During filtration through the membrane, the active binding site of the membrane adsorbs the target pollutants and efficiently removes contaminants from water.

There are many different methods of polymer synthesis for electrospinning, e.g., surface modification, graft polymerization, and blending. Moreover, various parameters can be adjusted during the electrospinning process to control the characteristics of the final membrane [15,16]. Nanofibers can be spun into structures of high porosity, small pore size, and high surface-area-to-volume ratio [17]. Flexible functionalization of an adsorptive electrospun nanofiber membrane leads to enhanced adsorption capacity and excellent thermomechanical properties [16,17].

Still, many challenges in water technology remain unresolved due to the complicated nature of NOM [7]. The effectiveness of NOM removal from water depends on the physicochemical properties of carbon, the quantity and quality of the pollutants, and the flow parameters.

The objectives of this study are to explore the application of PAN–PAC nanofibers without any post-modification to NOM removal. This study presents the results of bench-scale research on the efficiency of electrospun nanofibers in NOM removal. The adsorption of ENM, PAN virgin membrane, and PAN–PAC membrane were also compared.

## 2. Materials and Methods

### 2.1. Reagents and Water Source

Polyacrylonitrile (PAN) with an average molecular weight of 150 kDa and PAC (purity > 99%, DARCO^®^, ~100 mesh) were used in this study. The solvent for the electrospinning of the polymer blend was a reagent grade dimethylformamide (DMF). Solutions in DMF and an aqueous solution of sodium thiocyanate are the most widely used in the production of PAN fibers [18]. Wu et al. (2012) reported that the monomer of PAN interacts with each solvent through dipole–dipole interaction and forms PAN’-solvent complexes, and these complexes display an antiparallel alignment of an interacting pair between the C≡N group of PAN’ and the polar group of the solvent molecule [19].

All chemicals were reagent grade. Deionized (DI) water (MilliporeMilli-Q) was used to prepare the stock solutions.

Raw water was collected on several occasions from Lake Ontario. Before the actual filtration, the water was filtered through a paper filter to remove larger solid impurities. The water was at pH 6.94 and contained 2.2~3.1 mg/L of dissolved organic carbon (DOC), and its UV absorbance at 254 nm (UV254) was 0.036~0.011 1/cm. Lake Ontario has moderate to high levels of dissolved organic carbon (DOC) and total phosphorous concentrations between 4 and 12 μg/L [20].

Membrane filtration tests used dead-end filtration through a 47 mm polycarbonate holder with an effective filtration area of 9.6 cm² (PALL corporation). During the filtration, the transmembrane pressure (TMP) was constantly monitored electronically. The lake water was delivered to the membrane by a peristaltic pump at a constant flux of 150 L/m^2^/h. Permeate samples were collected for analysis of UV254. Prior to the introduction of lake water, the membrane was treated with DI for 24 h.

### 2.2. Membrane Fabrication

The polymer solution of 12% (PAN) in dimethylformamide (DMF) was heated overnight at 52 °C in an oven. After 24 h, the bottle was shaken manually for 5 min and then stirred for 24 h at room temperature to achieve homogeneity.

In a separate bottle, a 12 wt% suspension of powdered activated carbon (PAC) in DMF was prepared. ENM was prepared by mixing PAN and PAC in a 2:1 mass ratio and fabricating the ENM in a single step (Figure 1). PAN viscosity was 1.530 × 10^3^ mPa⋅s.

The employed electrospinning apparatus was a KH-1-1 type electrostatic spinning machine manufactured by Ji’nan Liang Rui Technology Co. (China). The polymer solution was loaded into a 20 mL syringe with an 18-gauge needle tip. The electrospinning process was executed with a flow rate of 90 mL/h. The applied voltage between the needle and the collector drum was 25 kV, and the distance between them was fixed at 20 cm. The electrospun nanofibers were collected by a metal drum collector covered with aluminum foil. Due to the limitations of the spinning machine, only one concentration of the addition could be spun. A similar membrane fabrication procedure was applied by Soberman et al. [21].

To evaluate the impact of incorporation of PAC nanoparticles, pure PAN membrane was also elecrospun with the application of the abovementioned procedure.

### 2.3. Filtration Tests

In the filtration experiments, the ENM membrane was placed in the cartridge unit, and the feed water from Lake Ontario (LO) was introduced (Figure 2). The feed water and permeate were tested after each filtration test. It was clear that the color of the PAN membrane after filtration had changed from white to yellow, suggesting the retention of some NOM from the feed water, whereas with PAN–PAC membrane, there was no clear change on the membrane surface (Figure 3).

### 2.4. Characterization of the Prepared Membrane

The morphology of the electospun nanofibers was examined by the field emission scanning electron microscope (FE-SEM) Hitachi SU5000 (Toronto, ON, Canada). ImageJ 1.53e software was used for the analysis of membrane porous scaffolds.

The FT-IR was obtained with the use of Perkin Elmer UATR Single Bounce on Diamond Crystal Pike Technologies VeeMax (Toronto, ON, Canada).

X-ray photoelectron spectroscopy (XPS) was acquired using ThermoFisher Scientific K-Alpha (Toronto, ON, Canada).

## 3. Results

The element analysis was used to evaluate NOM structure in samples of feed water and permeate collected after each filtration experiment and analyzed. The NOM removal efficiency was measured by UV254 nm, which is typically used for the evaluation of NOM’s aromatic compounds, and TOC, which is considered a quantitative measurement of the total organic carbon constituent of NOM.

The virgin PAN membrane contains fibers with an average diameter of 255 nm. The fibers are well layered with interconnected pores ranging from a few microns to a few tens of microns (Figure 4a–c). Large pores allow for the passage of water with minimum resistance.

After filtering the raw water through the PAN membrane, a few particles (foulants) can be seen attached to the fibers (Figure 4d–f).

The virgin PAN–PAC membrane had an average fiber diameter of 650 nm. Between fibers, the agglomeration of PAC particles can be observed (Figure 4g–i). With the addition of PAC particles, the fibers were transformed into irregular rough fibers. Most of the particles were embedded in PAN fibers and protruded from the fibrous surface. The PAC particles, when too large to be trapped within the fibers, were entangled with fiber knotting (Figure 4j–l). A similar observation was reported by Naseeb et al. (2019) in the case of the addition of silica nanoparticles to the PAN membrane, and it was ascribed to rapid evaporation of DMF [22]. After filtration, the surface of the PAN–PAC membrane is entirely covered with particles (foulants).

In order to evaluate the chemical change in the membrane surface, an FTIR was used (Figure 5). The FTIR results of PAN and PAN–PAC membranes are quite similar, except for peaks of variable intensity in the region of 1356 and 1072 cm^−1^. In the case of PAN–PAC membrane, the intensity of the peak of stretching vibration of nitrile group weakened. A pure PAN membrane consists of functional groups: methyl (CH_3_) and nitrile (C≡N). The C–H stretching band of methylene can be identified at 1453 cm^−1,^ and the peak at 2938 cm^−1^ belongs to the C–H stretching vibration. The vibration at 2242 cm^−1^ is commonly attributed to C≡N. Similar observation was made by Andrei et al. (2020) when evaluating the impact of carbonization on membrane fibers [23]. Additionally, they observed that during the carbonization treatment, C≡N triple bonds diminished [23]. Glass et al. (2021) had prepared amine-modified membranes and observed decreasing in the intensity of the nitrile band as well as increasing of the zeta potential of the membranes [6].

The most noticeable peaks in the XRD spectra are those corresponding to carbon, oxygen, and nitrogen (Figure 6). In the case of the virgin PAN–PAC membrane, the new peaks of S and Si are also visible. PAN–PAC–LO represents the membrane after filtration, where an increase in the relative peak areas of oxygen-containing groups was recorded. The increasing oxygen fraction in fouled membranes can be attributed to the presence of NOM on the membrane surface.

### Performance and Evaluation of the Developed Membranes in Natural Water Filtration

The developed membranes electrospun of pristine PAN, PAN–PAC were tested in a dead-end filtration mode. After the separation tests, the permeates were evaluated for the presence of natural organic matter measured by UV254 nm. The TMP increase was negligible on both PAN and PAN–PAC membranes (Figure 7). However, the rejection percentage of NOM was significantly higher in the case of the membrane with PAC nanoparticles.

Modification of polymer composition and especially a nanofiber load, may result in higher and faster contaminants removal [15,16,17]. Additionally, the functionalization of nanoparticles incorporated in the electrospun nanofibers can also radically improve the removal efficiency. Soberman et al. (2020) pointed out that salt addition (NaCL) to PAN–PAC polymer significantly improved the affinity of the membrane to methylene blue (MB) [21]. Moreover, Tajer et al. (2020) discovered that loading a high amount of activated carbon yields more available adsorption binding sites in the membranes [24]. Other tests affirmed dye adsorption improvement by incorporating nanofibers in the membrane mat [17,24,25,26,27,28,29,30,31,32,33].

The tested technique allows for a wide range of possibilities of modifying polymer blends while changing process parameters, allowing for the development of a material that can be more commonly used in industries.

## 4. Conclusions

The percentage removal efficiency measured by UV254 absorbance using only PAN membrane was up to 13%. In the case of PAN–PAC, the general rejection trend of NOM varied between 66% and 79%;SEM imaging exhibited different degrees and modes of coverage for PAC particles of different sizes when embedded in the membrane. FTIR spectra of both membranes are quite similar except for two peaks assigned to the carbonyl group and aromatic groups. XDR shows the presence of carbon, oxygen, and nitrogen groups in both membranes. After filtration, the increase of oxygen-containing groups was observed;The process can be studied further for high-performance membranes to provide low-cost treatment systems.

## Figures and Tables

**Figure 1 materials-14-04426-f001:**
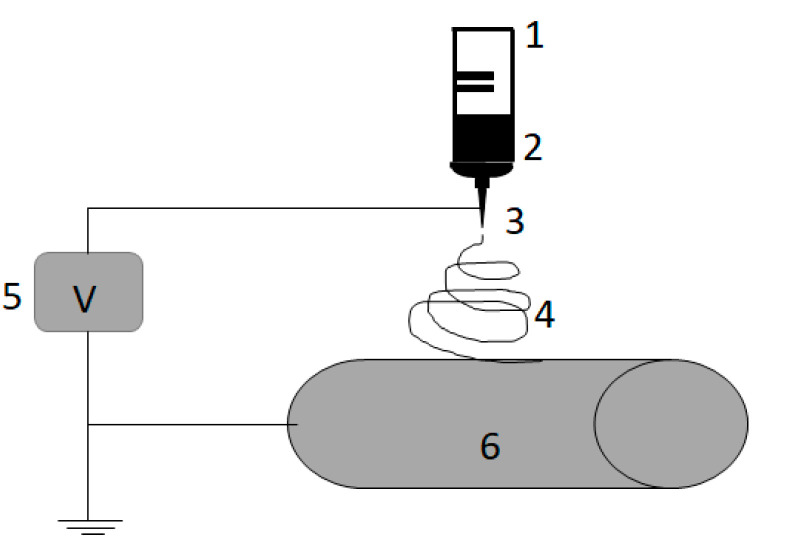
Schematic illustration of the electrospinning system: 1—syringe, 2—polymer solution, 3—needle, 4—liquid jet, 5—high voltage power supply, 6—collector.

**Figure 2 materials-14-04426-f002:**
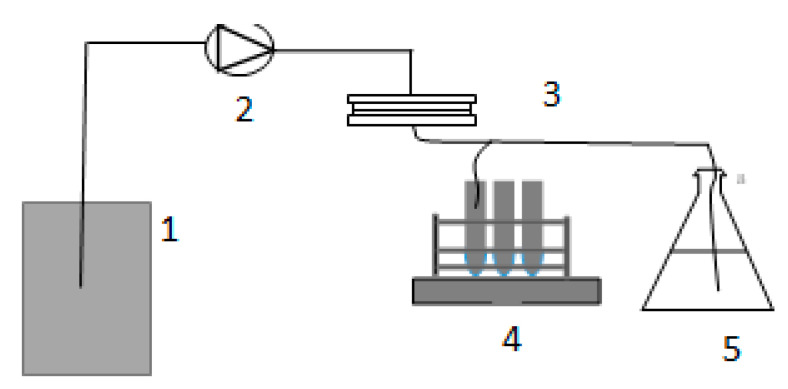
The bench-scale system used for the research: 1—feed water, 2—feed water pump, 3—membrane unit, 4—autosampler, 5—permeate tank.

**Figure 3 materials-14-04426-f003:**
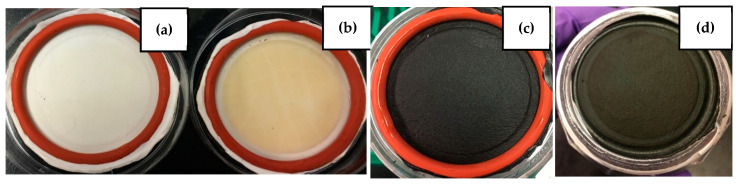
Membranes before and after filtration: (**a**)—PAN membrane, (**b**)—PAN membrane after LO filtration, (**c**)—PAN–PAC membrane prior to filtration, (**d**)—PAN–PAC after LO filtration.

**Figure 4 materials-14-04426-f004:**
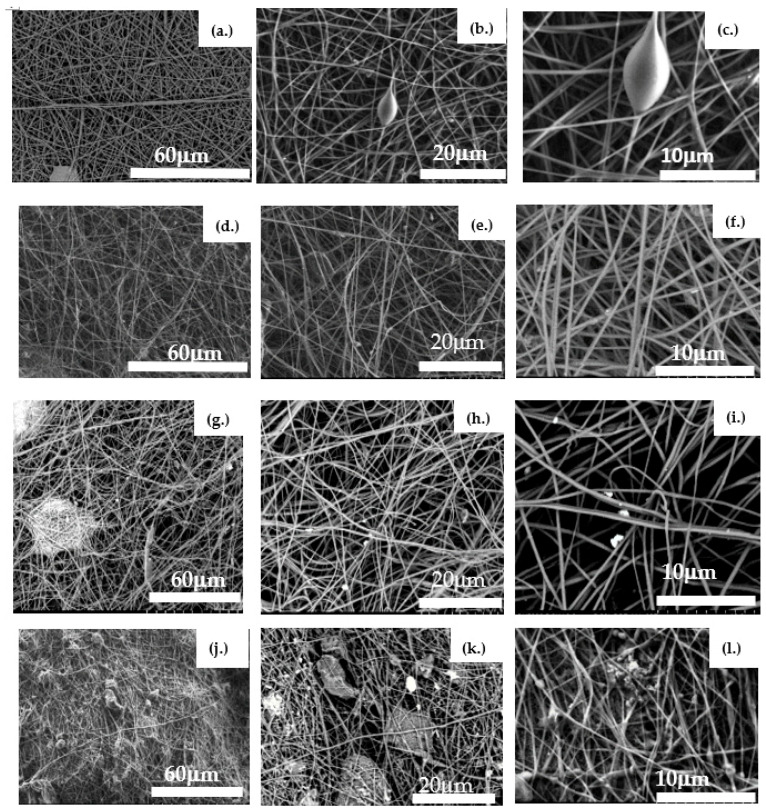
SEM images of the evaluated membranes. Images (**a**–**c**) depict a virgin PAN membrane before filtration; Images (**d**–**f**) show a virgin PAN membrane after filtering raw water from Lake Ontario; Images (**g**–**i**) present a virgin PAN–PAC membrane; and Images (**j**–**l**) represent a PAN–PAC membrane after filtration of raw water from Lake Ontario.

**Figure 5 materials-14-04426-f005:**
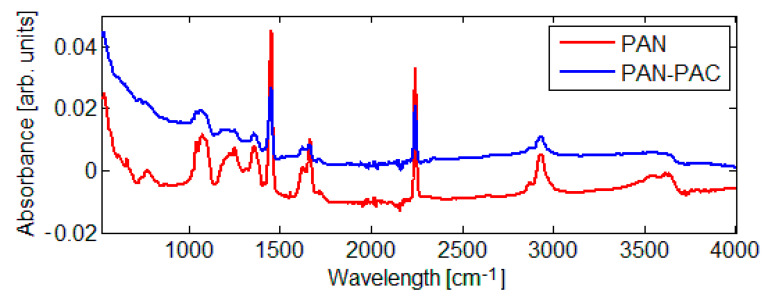
FTIR spectra of the evaluated membranes.

**Figure 6 materials-14-04426-f006:**
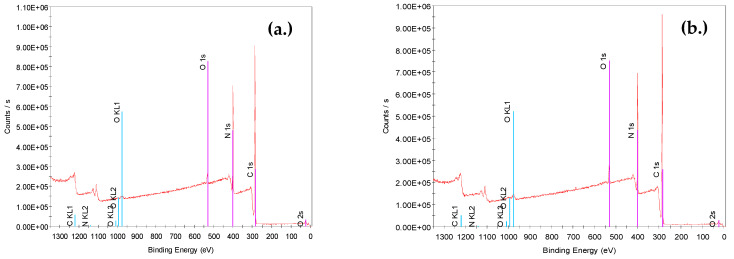
Images of XRD of (**a**) virgin PAN membrane, (**b**) PAN membrane after LO filtration, (**c**) virgin PAN–PAC membrane, and (**d**) PAC–PAC membrane after LO filtration.

**Figure 7 materials-14-04426-f007:**
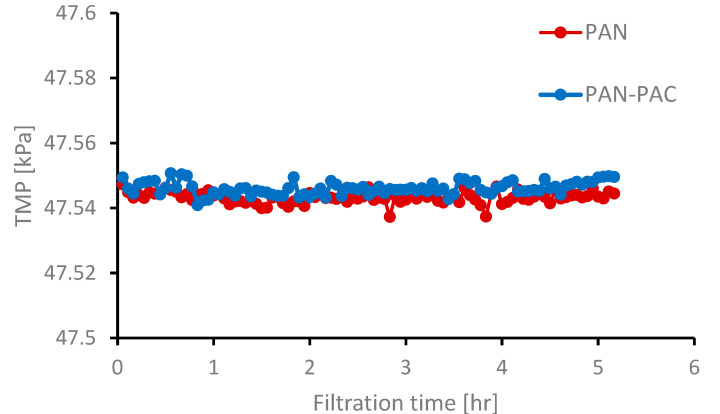
Transmembrane pressure (TMP) versus time.

## Data Availability

Not applicable.

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
