# Peer review of "Preparation and Performance of PAN–PAC Nanofibers by Electrospinning Process to Remove NOM from Water"

_materials, 2021, doi:10.3390/ma14164426_

Round 1

Reviewer 1 Report

The manuscript requires serious revision. It is necessary to deepen the theoretical component. The author must justify the novelty and relevance of the research. When describing the results, one should conduct a deeper analysis, compare the methods and conclusions, draw analogies and compare the results obtained with the literature data. 

1. I recommend shortening the title of the article.
2. "The water was at pH 6.94 and contained 2.2∼3.1 mg / L DOC" - it is necessary to decode "DOC"
3. What is the viscosity of the used PAN solution?
4. "In a separate bottle, a 12 wt% solution of powdered activated carbon (PAC) in DMF was prepared". Are you sure it's a PAC solution and not a suspension?
5. It is necessary to add some explanation to Figure 4.
6. Figure 5 you need to add a scale.
7. "The contact angle between PAN and PAN-PAC decreased by 98%. PAN membrane is hydrophilic unlike PAN-PAC." - Check this statement. 

Author Response

The author thanks the reviewers for their time and valuable comments.

Responses to reviewers’ comments

Thank you for your letter and the opportunity to revise the paper.

 - I recommend shortening the title of the article.

Response:  The manuscript has been reviewed and corrected as suggested. New title of the manuscript is: PREPARATION AND PERFORMANCE OF PAN-PAC NANOFIBERS BY ELECTROSPINNING PROCESS TO REMOVE NOM FROM WATER

-"The water was at pH 6.94 and contained 2.2∼3.1 mg / L DOC" - it is necessary to decode "DOC"

Response:  The manuscript has been reviewed and corrected.

What is the viscosity of the used PAN solution?

Response:  The manuscript has been reviewed and corrected. PAN viscosity was 1.530mPa×s.

In a separate bottle, a 12 wt% solution of powdered activated carbon (PAC) in DMF was prepared". Are you sure it's a PAC solution and not a suspension?

Response:  The manuscript has been reviewed and corrected: In a separate bottle, a 12 wt% suspension of powdered activated carbon (PAC) in DMF was prepared. 

It is necessary to add some explanation to Figure 4.

Response:  The manuscript has been reviewed and corrected.New table with contact angle details was provided.

 Figure 5 you need to add a scale.

Response:  The manuscript has been reviewed and corrected.

The contact angle between PAN and PAN-PAC decreased by 98%. PAN membrane is hydrophilic unlike PAN-PAC." - Check this statement. 

Response:  The manuscript has been reviewed and corrected.

Reviewer 2 Report

This communication provides an interesting insight into performance of a PAN membrane modified with powdered activated carbon using electrospinning. In
general, the narrative is well written although in places there are spelling mistakes and the language could be tightened up somewhat. Below I include my comments, if the author can add the necessary details indicated below then I feel that this study is ultimately worthy of publication.

Introduction

  1. It is stated that “AC modifications have been attracting more attention in attempts to increase its affinity to target contaminants in order to improve efficiency of their removal”. AC is highly efficient adsorptive material for organic materials. The author need to specify modification with respect to contaminants of other nature. Moreover, “AC modifications have been attracting more attention in attempts to increase its affinity to target contaminants in order to improve efficiency of their removal. Therefore, there are many approaches to the fabrication of various carbon-based materials by physical, chemical, and biological modifications of AC characteristics” is irrelevant as this communication deals with use of unmodified AC.
  2. Reference for “AC adsorption has proven to be effective for the removal of several types of inorganic and organic contaminants both as a stand-alone step and in combination with other conventional and advanced water and wastewater treatment systems [2].” seems to be incorrect or insufficient. Please provide the reference relevant for the discussion.
  3. It is stated that “depending on the pore size, it is possible to remove contaminants at an ionic level.” The surface charge of the membrane is exceedingly important and plays a deceive role in membrane processes.
  4. Reference 28 is incomplete.
  5. Knowledge gap is unclear in the introduction. It is unclear why the electrospinning techniques for fabrication of mixed matrix membrane should be preferred. To control membrane fouling and expand membrane performance for different contaminants and, dynamic membranes have shown quite promising outcomes. Comparing the performance of the dynamic membrane, it will be easy for the readers to determine the novelty of this work.
  6. Reference is needed for “Still, many challenges in water technology remain unresolved due to the complicated nature of NOM. The effectiveness of NOM removal from water depends on the physicochemical properties of carbon, the quantity and quality of the pollutants, and the flow parameters.”
  7. The objective of the work shall be written more precisely.

Materials and methods

  1. It is important to add more details on the source of surface water

Results

  1. 6. It is difficult to differentiate between the FTIR spectra’s of PAN and PAC-PAC. Moreover, PAN membrane show a nitrile functional group at 2250 cm-1. The author shall compare the FTIR spectra of PAN membrane with literature (for example 10.1002/cite.202100037) to compare the functional groups of the PAN membranes that are necessary for modification of the PAN membranes.
  2. It will be excellent to add a graph on results of PAN and modified PAN membrane demonstrating the NOM retention/adsorption.

Author Response

Thank you for your letter and the opportunity to revise the paper.

AC is highly efficient adsorptive material for organic materials. Often used to remove contaminants that cause membrane fouling. In different literature reports it has different efficiencies in membrane protection.

This paper foused on application of electrospinning technique for fabrication membrane. In priory study, an ENM was prepared by mixing polyacrylonitrile (PAN), a hydrophilic polymer, and PAC in a 2:1 mass ratio and fabricating the ENM in a single step. No post-modification strategy was applied. This resulted in a mechanically stable PAC-ENM that successfully adsorbed the NOM from water.

It is stated that “depending on the pore size, it is possible to remove contaminants at an ionic level.” The surface charge of the membrane is exceedingly important and plays a deceive role in membrane processes.

In the water industry, an ultrafiltration membrane is less effective than RO, which removes contaminants from water at the ionic level. This is a general statement. I agree that surface charge is important but as well as water composition (e.g. ionic charge).  Additionally, if the ampholytic polymer contains weak basic and/or weak acidic groups, the charge of the membranes is affected by the pH. These features in themselves require detailed discussion and were not covered in this article. 

The reference 28 was changed and revised as suggested.

More details on the source of surface water was also provided.

FTIR spectra of PAN membrane have been compared, but the suggested literature uses a completely different method of membrane fabrication. Therefore, the results may not be identical. 

The authors do not quite understand what kind of graph the reviewer expects when asking for the graph on results of PAN and modified PAN membrane demonstrating the NOM retention/adsorption? The percentage removal efficiency measured by UV254 absorbance using only PAN membrane was up to 13%. In the case of PAN-PAC, the general rejection trend of NOM varied between 66% up to 79%. This removal was calculated based on the composite permeate after the filtration. In the filtration experiment flux was fixed. The lake water was delivered to the membrane by a peristaltic pump at a constant flux of 150 L/m2-hr. The graph of TMP versus time was was added. 

Round 2

Reviewer 1 Report

The author made a number of edits to the manuscript, but in my opinion the work requires further revision. In addition to listing the results, it is necessary to deepen the discussion of the results. It is necessary to more accurately arrange the manuscript.

Author response to report 1:

What is the viscosity of the used PAN solution?

Response: The manuscript has been reviewed and corrected. PAN viscosity was 1.530mPa*s. 

I have a lot of doubt about this value. Check it please!

Why DMF was chosen as a solvent. You can take, for example, NMMO and prepare a 12% solution in a short period of time (Golova LK et al. Peculiarities of Dissolving Polyacrylonitrile Copolymer Containing Methylsulfo Groups in N-Methylmorpholine-N-Oxide. Polym. Sci. Ser. A 62, 597– 606 (2020). Https://doi.org/10.1134/S0965545X20060036). Rationale for solvent selection DMF and other direct PAN solvents should be added to the work.

It is desirable to change this proposal "The FTIR results of PAN and PAN-PAC membranes are mostly the same, except for a peak observed at 1995 cm − 1 in the spectrum" or remove "membranes are mostly the same"

Author Response

The author thanks the reviewers for their time and valuable comments.

The changes to improve the clarity have been made throughout the manuscript, and in the sections on interpretation.

The viscosity of PAN solution was measured. The measurements have been carried out using VT550 viscometer (manufactured by Haake) with rotating coaxial cylinder. PAN viscosity was 1.530*103mPa×s.

The DMF was chosen as a solvent because it is the most widely used in production of PAN fibres. Solutions in DMF and aqueous solution of sodium thiocyanate are the most widely used in production of PAN fibers [18]. Wu et al. (2012) reported that the monomer of PAN interacts with each solvent through dipole-dipole interaction and forms PAN'-solvent complexes and these complexes display an antiparallel alignment of an interacting pair between the C≡N group of PAN’ and the polar group of solvent molecule [19].

The FTIR text has been changed to correct this.
